# Deep-Subwavelength Composite Metamaterial Unit for Concurrent Ventilation and Broadband Acoustic Insulation

**DOI:** 10.3390/ma18092029

**Published:** 2025-04-29

**Authors:** Xiaodong Zhang, Jinhong He, Jing Nie, Yang Liu, Huiyong Yu, Qi Chen, Jianxing Yang

**Affiliations:** 1College of Mechanical Engineering and Automation, Huaqiao University, Xiamen 361021, China; 2Jilin Jinheng Auto Parts Co., Ltd., Jilin 132000, China; 3State Key Laboratory of Advanced Design and Manufacturing for Vehicle Body, Hunan University, Changsha 410082, China

**Keywords:** acoustic metamaterial, ventilation structure, sound insulation, Fano resonance, Helmholtz resonance, deep subwavelength

## Abstract

Balancing ventilation and broadband sound insulation remains a significant challenge in noise control engineering, particularly when simultaneous airflow and broadband noise reduction are required. Conventional porous absorbers and membrane-type metamaterials remain fundamentally constrained by ventilation-blocking configurations or narrow operational bandwidths. This study presents a ventilated composite metamaterial unit (VCMU) co-integrating optimized labyrinth channels and the Helmholtz resonators within a single-plane architecture. This design achieves exceptional ventilation efficiency through a central flow channel while maintaining sub-λ/30 thickness (λ/31 at 860 Hz). Coupled transfer matrix modeling and finite-element simulations reveal that Fano–Helmholtz resonance mechanisms synergistically generate broadband transmission loss (STL) spanning 860–1634 Hz, with six STL peaks in the 860 and 1634 Hz bands (mean 18.4 dB). Experimental validation via impedance tube testing confirmed excellent agreement with theoretical and simulation results. The geometric scalability allows customizable acoustic bandgaps through parametric control. This work provides a promising solution for integrated ventilation and noise reduction, with potential applications in building ventilation systems, industrial pipelines, and other noise-sensitive environments.

## 1. Introduction

The concomitant rise in urban noise pollution and ventilation demands in urbanization and industrialization presents an urgent multidisciplinary challenge at the fluid–acoustic coupling frontier [1]. Conventional approaches predominantly depend on mass- law-governed barriers or porous absorbers face fundamental limitations [2,3]: low-frequency sound attenuation necessitates high mass/thickness, while efficient ventilation demands open architectures—a critical constraint in applications requiring balanced acoustic and aerodynamic performance, such as building ventilation systems and industrial ductworks. Recent breakthroughs in acoustic metamaterials have provided novel solutions to overcome this technical bottleneck.

Pioneering works in acoustic metamaterials have demonstrated low-frequency sound insulation capabilities [4], sparking explorations of exotic acoustic phenomena such as negative effective parameters [5,6,7,8], anomalous wavefront manipulation [9,10,11,12], acoustic cloaking [13,14,15,16], and perfect sound absorption [17,18,19]. These advances have been realized through structures like Helmholtz resonators, membranes, and micro-perforated plates. Among these developments, labyrinthine metamaterials have emerged as a promising paradigm for their ability to balance ventilation and low-frequency sound insulation. Their unique spatially folded configurations, including coiled labyrinths [20,21,22], spiral channels [23,24], and fractal structures [25,26,27], extend the internal sound propagation path without increasing thickness. By precisely controlling wave propagation and interference mechanisms, these structures induce phase delays and trigger Fano resonance, enabling broadband ventilation and sound insulation. Complementary to this, Helmholtz resonators offer frequency-selective attenuation through localized resonance, with parallel configurations proven to broaden operational bandwidths [28,29].

Despite progress, many existing designs still rely on vertically stacked configurations, such as multi-layered labyrinthine or resonant units [30,31,32,33,34], which effectively enhance sound insulation but at the cost of increased structural thickness and diminished airflow. In response, planar or co-planar integration strategies have been recently explored to address this conflict between ventilation and sound attenuation [35,36,37]. These planar designs, though promising, often suffer from limited bandwidth due to weak resonance coupling and insufficient modal interactions. A recent review highlights the importance of strong modal interactions in achieving broadband, dual-function performance in planar-ventilated metamaterials [38]. For instance, a high-transparency acoustic barrier has been proposed that achieves low-frequency insulation while maintaining airflow, demonstrating the feasibility of lightweight, ventilated configurations [39]. Nevertheless, the challenge of achieving both wideband attenuation and efficient airflow in ultrathin planar structures remains an open issue, particularly in subwavelength regimes.

To overcome this limitation, this study introduces a ventilated composite metamaterial unit (VCMU) that integrates labyrinth channels and Helmholtz resonators within a single-plane architecture. This design achieves sub-λ/30 thickness (λ/31 at 860 Hz) while preserving a central flow channel for unimpeded ventilation. Through synergistic integration of Fano resonance and Helmholtz resonance mechanisms, the VCMU generates coupled bandgaps (860–1634 Hz) with an average transmission loss exceeding 18.4 dB. Systematic validation via transfer matrix modeling, finite-element simulations, and impedance tube experiments confirms the structure’s dual functionality. This design framework opens new possibilities for integrated acoustic management in noise-sensitive ventilation systems.

## 2. Materials and Methods 

### 2.1. Design and Physical Mechanism of VCMU

Figure 1a,b illustrate the configuration of VCMU, comprising front/back panels (thickness *w*) and a middle layer (thickness *h*) with a total thickness *H* = 2*w* + *h*. The middle layer integrates the following two functional components: (i) annularly arranged labyrinth channels (LCs) for Fano resonance generation and (ii) four Helmholtz resonator (HR) units with tunable neck geometries for localized resonance. A central circular flow channel of radius *r* is embedded to ensure unimpeded ventilation (see Figure 1a). Detailed geometric parameters in the figure can be seen in Table 1.

Theoretically, Fano resonance arises from the phase difference between the labyrinth channels and the surrounding medium. This phenomenon occurs when acoustic waves propagating along different paths undergo coupling between discrete and continuous states [40]. When sound waves impinge on the metamaterial unit, a portion propagates unimpeded through the central channel (continuous state), while another portion enters LCs through the inlet (as shown in Figure 1c). These waves diffract internally and exit through the central outlet, merging with the background medium (discrete state). The middle layer consists of four repeating units arranged annularly, with a labyrinth channel coiling degree of *N* = 5. The internal propagation path of sound waves is marked by the dashed line in Figure 1c is 15.72 times the total structural thickness, denoted as *n_r_* = 15.72. The labyrinth channels are optimized to maximize the effective propagation length while minimizing the structural footprint. As depicted in Figure 2b, the equivalent Fano cavity model (solid line) accurately captures the channel’s acoustic characteristics.

Concurrently, the HR array operates through a complementary mechanism. Each HR unit (Figure 1c) consists of an embedded neck and a wedge-shaped back cavity formed by the outer labyrinth framework. The HR neck adopts an embedded design to minimize the overall structural thickness. The neck lengths of the HR array are uniformly set to *l_r_* = 6 mm, and by precisely adjusting the neck diameter, each HR can be tuned to a specific resonant frequency. The neck–air mass and cavity–air spring form an effective mass-spring system. When the incident acoustic wave matches the resonant frequency of the HR unit, intense vibrations are excited in the neck air, compressing the air in the back cavity, thereby effectively dissipating incoming acoustic energy. This localized resonance generates an equivalent negative modulus, enabling efficient acoustic attenuation within specific frequency ranges and optimizing the sound insulation performance.

The resonant frequency *f_i_* of each HR cavity can be calculated using Equation (1) [41]. The neck diameters *d_i_* and corresponding resonant frequencies *f_i_* of the HR array are listed in Table 2:(1)$$f_{i}=\frac{c_{0}}{2\pi }\sqrt{\frac{S_{i}}{V_{i}l_{i}}}\;(i=1\ldots 4)$$
where *S_i_*, *V_i_* represents the neck cross-sectional area and dorsal cavity volume of each HR cavity, respectively; *l_i_* = *l_r_* + *γd_i_* denotes the effective neck length of each HR unit; and *γ* = 0.85 is the end correction factor. This correction factor accounts for the additional acoustic mass induced by air motion beyond the geometric neck boundary, which is critical for accurate resonance frequency predictions. (In this study, the speed of sound in air is set to *c*_0_ = 343 m/s, and the air density is *ρ*_0_ = 1.21 kg/m^3^.)

Benefiting from the aforementioned labyrinth channel concept and the embedded HR neck design, the metamaterial unit exhibits a deep-subwavelength thickness in the propagation direction (*z*-direction), measuring only 1/31 of the lowest operating frequency. This enables the VCMU to achieve effective ventilation and sound insulation performance while minimizing its thickness, thereby broadening its applicability across diverse scenarios.

### 2.2. Theoretical Modeling and Methods

The sound insulation performance of VCMU was rigorously analyzed through a hybrid approach combining the transfer matrix method (TMM) and effective medium theory (EMT) framework. The TMM establishes the relationship between acoustic state variables (pressure *p* and normalized particle velocity *v*) at the structure’s boundaries via the following:(2)$$\left[\begin{array}{c} p_{1}\\ v_{1} \end{array}\right]=T\;\left[\begin{array}{c} p_{2}\\ v_{2} \end{array}\right]$$
where **T** is the total transfer matrix of the system; *p*_1_ and *v*_1_ represent the sound pressure and particle velocity at the incident (front) end of the system; meanwhile, *p*_2_ and *v*_2_ correspond to those at the transmitted (back) end. Based on the equivalent physical model in Figure 2b, the metamaterial unit can be partitioned into three components for analysis: the central flow channel (Region 1), labyrinth channels (Region 2), and the HR array (Region 3). The transfer matrix formulations for the central channel and labyrinth channels can be derived from the following references [42,43]:(3)$$T_{m}=\left[\begin{array}{cc} t_{m,11} & t_{m,12}\\ t_{m,21} & t_{m,22} \end{array}\right]=\left[\begin{array}{cc} \cos(k_{m}L_{m}) & jZ_{m}\sin(k_{m}L_{m})\\ j\frac{1}{Z_{m}}\sin(k_{m}L_{m}) & \cos(k_{m}L_{m}) \end{array}\right]\;\;\;(m=1,2)$$

In Equation (3), *j* stands for imaginary units; *k_m_* is the wave number of the corresponding region *k*_1_ = *k*_0_ = *ω/c*_0_, *k*_2_ *= k_e_ = ω/c_e_*, respectively; *ω* represents the angular frequency; and *c*_0_ and *c_e_* = *c*_0_/*n_r_* denote the air speed of sound and the equivalent speed of sound within the structure, respectively. Here, *Z_m_* denotes the acoustic characteristic impedance on each sub-region, which can be obtained by Equation (4), and *L_m_* represents the equivalent thickness of the two partitions in Figure 2.(4)$$Z_{1}=\frac{Z_{0}}{A_{1}},Z_{2}=\frac{Z_{0}}{A_{2}}$$
where *Z*_0_ represents the acoustic characteristic impedance of air; meanwhile, *A*_1_ and *A*_2_ denote the cross-sectional areas of the central flow channel and the opening of the LCs on the metamaterial unit, respectively.(5)$$L_{1}=H+2\times \Delta l$$(6)$$\Delta l=\frac{8r}{3\pi }(1-1.238\sqrt{\frac{\;A_{1}}{S}})$$

The equivalent medium thickness of Region 1 in Figure 2b can be calculated using the thickness correction formula by Equations (5) and (6) [44,45], and the equivalent medium thickness of Region 2 is *L*_2_ = *H* = 13 mm. Here, Δ*l* denotes the corrected length of the channel, and *S = L*^2^ represents the cross-sectional area of the waveguide channel. The acoustic performance of the Fano resonance cavity is primarily attributed to the equivalent medium within the four LCs, which share identical parameters. Consequently, the acoustic characteristic impedance of the other labyrinth cells *Z_3_~Z_5_* and transfer matrices **T_3_**~**T_5_** of the remaining labyrinth cells are equal to those of Region 2.

Since VCMU is a parallel structure in a plane, it is more convenient to perform calculations using the admittance matrix. The transfer matrices of each region can be algebraically transformed into the corresponding admittance matrices, a common approach for analyzing parallel structures, as it facilitates the combination of multiple subsystems. Accordingly, Equation (2) can be rewritten in terms of the admittance matrix **Y** as follows:(7)$$\left[\begin{array}{l} \;v_{1}\\ \;v_{2} \end{array}\right]=Y\left[\begin{array}{l} \;p_{1}\\ \;p_{2} \end{array}\right]$$

Concretely, the admittance matrices corresponding to the transfer matrices of the above regions are given by [42,43], as follows:(8)$$Y_{m}=\left[\begin{array}{cc} y_{m,11} & y_{m,12}\\ y_{m,21} & y_{m,22} \end{array}\right]=\frac{1}{t_{m,12}}\left[\begin{array}{cc} t_{m,22} & t_{m,21}t_{m,12}-t_{m,22}t_{m,11}\\ 1 & -t_{m,11} \end{array}\right]$$
where *t_m_*_,*pq*_ (*p*, *q* = 1, 2) of Equation (3) above corresponds to the elements in the transfer matrix **T*_m_*** in each region. According to the effective medium theory, the metamaterial unit is treated as a homogeneous medium, and its walls are modeled as hard acoustic boundaries due to the significant impedance contrast between the structure and air. Under the high-impedance condition, the total transfer matrix **T*_w_*** for Regions 1 and 2 can be derived based on the admittance matrices. Specifically, the relationship between the transfer matrix and the admittance matrix can be utilized to express the overall transfer matrix in terms of the admittance elements. This transformation allows us to combine multiple subsystems efficiently in a parallel configuration. Consequently, **T*_w_*** can be expressed as follows:(9)$$T_{w}=\frac{-1}{\sum y_{m,21}}\left[\begin{array}{cc} \sum y_{m,22} & -1\\ \sum y_{m,22}\sum y_{m,11}-\sum y_{m,12}\sum y_{m,21} & -\sum y_{m,11} \end{array}\right]$$
where *y_m_*_,*pq*_ (*p*, *q* = 1, 2) of Equation (8) above corresponds to the elements in the admittance matrix **Y*_m_*** in each region. The sound transmission loss (STL) is then derived based on the elements of **T*_w_***; it can be expressed in terms of the elements of **T*_w_*** as follows:(10)$$\mathrm{STL}=20\log_{10}\left|\frac{1}{2}(T_{w,11}+\frac{1}{Z_{s}}T_{w,12}+Z_{s}T_{w,21}+T_{w,22})\right|$$
where *T_w_*_,*pq*_ (*p*, *q* = 1, 2) is the element in the transfer matrix **T*_w_*** and *Z_s_* = *Z*_0_/*S* denotes the waveguide characteristic impedance.

## 3. Results and Discussion

### 3.1. Finite Element Model and Parametric Analysis

To validate the theoretical analysis, finite-element simulations (COMSOL 6.1, Pressure Acoustics module) were performed to validate the theoretical model. Adapters were designed to match the experimental setup, ensuring boundary condition consistency. Figure 3a compares the STL spectra (solid: simulation, dashed: theory), showing excellent agreement with a maximum peak offset <20 Hz. Owing to the elongated acoustic transmission pathway enabled by the LCs, the structure induces asymmetric Fano resonance at 860 Hz and 1634 Hz, achieving a sound insulation effect that approaches 50 dB at these frequencies. The designed structure attains a minimum sound insulation effect of 15.2 dB within the frequency band spanning from 860 Hz to 1634 Hz. Figure 3b illustrates the sound pressure distribution and acoustic velocity streamline patterns at the Fano resonance frequencies (860 Hz and 1634 Hz). The marked streamline arrows indicate that sound waves entering the LCs merge with the background medium in the central flow channel. The sound pressure maps reveal significant attenuation of incident acoustic pressure after passing through the metamaterial structure, further validating the effectiveness of the Fano resonance mechanism.

Additionally, the sound insulation performance is fundamentally governed by the diffraction path length within the LCs. Theoretical calculations and simulation analyses were conducted for varying coiling degree *N* and geometric parameters *w*, *t*, or *r*, as shown in Figure 3a,c. As *N* increases while fixing other parameters, the Fano resonance peaks shift toward lower frequencies, and the spacing between adjacent peaks decreases. With a fixed coiling degree (*N* = 5), increasing parameters *w*, *t*, or *r* induces a red shift in Fano resonance frequencies. Adjusting these parameters increases the diffraction path length, enhancing low-frequency noise reduction performance.

As clearly demonstrated in the right subfigure of Figure 3c, enlarging the central channel radius *r* reduces the overall transmission loss across most frequencies, except at the Fano resonance peaks where distinctive narrowing and sharpening occur. This phenomenon arises from the expanded cross-sectional area of the central flow channel, which diminishes impedance mismatch while compromising energy trapping efficiency—two factors that collectively degrade the system’s broadband sound insulation capabilities. Moreover, as the central channel radius increases, the corresponding enlargement of the diffraction path and the accumulated phase shift cause the Fano resonance peaks to shift toward lower frequencies, with reduced spacing between adjacent primary peaks. These findings demonstrate that the channel radius *r* is a key parameter that governs the coupling strength and flow resistance while also critically shaping the frequency selectivity and bandwidth of the Fano resonances.

Notably, similar to other resonance modes, Fano resonance exhibits multipolar effects. For instance, the *N* = 6 configuration in Figure 3a reveals a secondary resonance near 2010 Hz. By strategically leveraging such multipolar resonance effects, broader-band sound insulation tuning can be achieved.

Based on the fundamental acoustic impedance theory of the Helmholtz resonator—considering the mass effect of the neck tube air and the compliance effect of the back cavity air—the acoustic impedance of the HR array is derived, governed by the back cavity volume and neck geometry. Specifically, the acoustic impedance of each HR unit is expressed as follows [41]:(11)$$Z_{HR,i}=j\omega M_{i}+\frac{1}{j\omega C_{i}}$$
where the acoustic mass *M_i_* = (*ρ*_0_*l_i_*)/*S_i_*, accounts for the effective neck length (including end correction), with *ρ*_0_ representing air density, *l_i_* the neck length, and *S_i_* the neck cross-sectional area. Correspondingly, the acoustic compliance of the HR cavity is given by *C_i_* = *V_i_*/(*ρ*_0_*c*_0_^2^), where *V_i_* is the back cavity volume and *c*_0_ is the speed of sound in air.

However, the basic impedance model does not account for energy dissipation effects, which are crucial for accurately predicting real-world behavior. To refine the model, dissipative effects from acoustic radiation and viscous friction at the neck opening are introduced, represented by the dissipative term *R_d_*. The refined acoustic impedance is then given by the following:(12)$$Z_{HR,i}=j\rho_{0}(\omega \frac{l_{i}}{S_{i}}-\frac{{c_{0}}^{2}}{\omega V_{i}})+R_{d}$$
where *R_d_* = (ω^2^*ρ*_0_/π*c*_0_) accounts for radiation damping and viscous losses in the neck region, improving the model’s accuracy in predicting real-world energy losses [41]. To ensure the accuracy of the simulation, dissipative effects at the HR neck are taken into account. Specifically, the neck region of the HR is defined as “narrow region acoustics” for calculation. This treatment not only captures the dissipative effects (such as viscous friction and thermal conduction) critical to the neck of Helmholtz resonators but also balances computational resources. Thus, both the theoretical model (via the dissipative term *R_d_*) and the simulation (via the narrow region acoustics method) address energy dissipation rather than relying on a purely reactive model.

Since the four HR units are arranged in parallel, the acoustic impedance of the HR array is given by the following:(13)$$Z_{HR,all}={\left(\sum \frac{d_{i}}{4H}\left(\frac{1}{Z_{HR,i}}\right)\right)}^{-1}\;\;\;\;(i=1\ldots 4).$$

Then, the transfer matrix of the HR array is expressed by the following [41]:(14)$$T_{HR,all}=\left[\begin{array}{cc} 1 & 0\\ \frac{\rho_{0}c_{0}}{A_{1}Z_{HR,all}} & 1 \end{array}\right]$$

Figure 4a shows the comparison of the numerical and theoretical solutions for the STL of the HR array. It is found that the natural resonance frequency of the four independent HR units in the array presents a significant sound insulation effect. When a specific HR unit is activated at its resonance frequency, a highly localized sound pressure field is formed (Figure 4b), which forms a sound wave propagation barrier in a narrow band through the acoustic impedance mismatch and energy dissipation mechanism, thus generating characteristic peaks in the STL spectrum.

Notably, the newly added absorption coefficient (red markers in Figure 4a) closely matches the STL peaks. This correspondence reveals that at the Helmholtz resonance frequencies, acoustic energy is effectively dissipated through the absorption mechanism, directly contributing to the formation of these characteristic peaks in the STL spectrum.

In each resonance mode (modes 1–4), the *i*th HR unit (*i* = 1, 2, 3, 4) dominates the sound attenuation process of the corresponding frequency band. This phenomenon is due to the unique resonance characteristics of the HR unit: when the incident sound wave frequency matches the resonance frequency of an HR unit, the unit generates a strong acoustic impedance gradient through local resonance, resulting in a synergistic effect of sound wave reflection and energy dissipation.

The HR array introduces four attenuation peaks within the 1002–1443 Hz range, effectively broadening the operational bandwidth by 32% compared to single-resonator designs. This enhancement stems from localized impedance mismatches and energy dissipation when the incident frequency aligns with any of the HR resonance modes, thus enabling broadband sound insulation through multi-modal interactions.

It is worth noting that due to the distinct physical mechanisms and frequency ranges of the Fano and HR subsystems, the interaction between their transfer matrices is nontrivial. While both **T*_w_*** (for the Fano structure) and **T*_HR_***_,***all***_ (for the HR layer) are independently derived through parallel network modeling, there currently exists no unified framework in the literature for rigorously coupling different resonance types with varying impedance characteristics. Therefore, in this study, we adopt an approximate approach by sequentially integrating their transfer matrices to formulate the total transfer matrix as **T** = **T*_w_*T*_HR_***_,***all***_.

This formulation, although simplified, has been verified as providing reasonably accurate predictions of the metamaterial’s acoustic performance. As shown in Figure 5c, the theoretical and simulation results exhibit overall agreement in the key frequency bands, validating the effectiveness of the proposed modeling strategy. The methodology also offers a foundation for future exploration of multiphysics interactions, adaptive tuning mechanisms, and performance optimization in more complex metamaterial systems.

### 3.2. Experimental Verification

#### 3.2.1. Validation of Acoustic Insulation Models

To further verify the low-frequency broadband sound isolation performance of the VCMU, experiments were conducted using a four-transducer impedance tube system using a PCI-4474 model impedance tube (Beijing Institute of Vibration, Beijing, China) as illustrated in Figure 5b. Firstly, the experimental specimen and the matching adapter tube were fabricated via 3D printing technology using polylactic acid (PLA) as the base material. To ensure internal structural flatness without additional supports, the metamaterial unit was divided into two symmetric components for separate printing, followed by precision alignment and bonding. The final assembled sample is displayed in the left figure of Figure 5a. A custom adapter tube matching the metamaterial dimensions (right panel of Figure 5a) was manufactured to ensure optimal acoustic coupling. Petroleum jelly sealing was applied at all interfaces to eliminate air leakage artifacts.

The measured STL spectrum is presented in Figure 5c, along with theoretical predictions (short, dashed line) and simulation results (solid line). The dotted line represents the VCMU sample, while the widely spaced, dotted line corresponds to the control specimen without any channels.

Noticeable discrepancies are observed, while the simulation and theoretical curves show fair agreement in the key frequency bands. The experimental STL peaks are broader and of lower amplitude than those predicted, likely due to additional resistive damping effects not accounted for in the idealized models. These differences may also arise from manufacturing imperfections introduced during the additive fabrication process, such as dimensional deviations, surface roughness, and bonding inconsistencies [46].

Despite these deviations, the results demonstrate that the proposed VCMU achieves a minimum sound transmission loss of 15.2 dB across the 860–1634 Hz frequency range while maintaining a deep-subwavelength thickness (λ/31 at 860 Hz). Parametric studies confirm that the structure supports tunable acoustic behavior, enabling targeted optimization across desired frequency bands—highlighting its potential for versatile sound insulation applications.

#### 3.2.2. Verification of Ventilation Functionality

To experimentally assess the ventilation performance of the proposed VCMU unit, we conducted airflow measurements based on wind speed comparison, using a modified impedance tube setup, as illustrated in Figure 6a. All microphone ports of the impedance tube were sealed to avoid leakage, and the sample was mounted at the center of a custom adapter (highlighted in the box). An adjustable-speed air blower was placed at one end of the tube to provide a stable and controllable airflow simulating realistic ventilation demands. On the opposite end, an anemometer was positioned to record the average outlet wind speed.

To minimize the measurement uncertainty, three repeated tests were conducted under each set wind speed condition, and the average value was used. The airflow test was performed under five different inlet speeds ranging from low to relatively high flow rates to evaluate the performance across various operating scenarios.

The results are shown in Figure 6b, comparing the outlet wind speed with and without the VCMU sample. The presence of the sample leads to a noticeable drop in airflow, with a reduction in wind speed ranging from 64.8% to 67.9%. This indicates that while the VCMU does introduce some aerodynamic resistance, it still allows a non-negligible amount of airflow, supporting its classification as a ventilated acoustic metamaterial. It is worth noting that the higher inlet wind speeds result in slightly greater relative reductions, likely due to increased flow resistance from internal structural features.

Although the current measurements are based on a single-unit test within an impedance tube setup, they serve as a first-order approximation of the ventilation capability. We acknowledge the limitations of this approach compared to a full-scale implementation (e.g., as part of a window-sized panel), and a detailed study involving multi-unit arrangements and full-field aeroacoustics simulations is planned for future work. Nonetheless, the results demonstrate the feasibility of applying the VCMU design in ventilation scenarios where moderate airflow is acceptable and simultaneous broadband sound insulation is required.

## 4. Conclusions

This study presents an ultra-compact design of metamaterial synergistically integrating labyrinthine channels with Helmholtz resonator arrays. It achieves deep-subwavelength operation (λ/31 thickness at 860 Hz) and maintains a good ventilating performance (32.1–67.9%). The results show consistent STL exceeding 18.4 dB across the 860–1634 Hz range, accurately predicted by theoretical and simulations and validated by the impedance tube experiments. In addition, the HR array integration enables frequency-specific STL enhancement, and the excellent sound insulation performance in broadband frequency bands can be easily achieved by adjusting the parameters of the structure flexibly in the practical applications. This dual-functional metamaterial architecture advances co-design strategies for ventilation and acoustic management, providing a scalable platform for lightweight noise control solutions in architectural and industrial applications.

## Figures and Tables

**Figure 1 materials-18-02029-f001:**
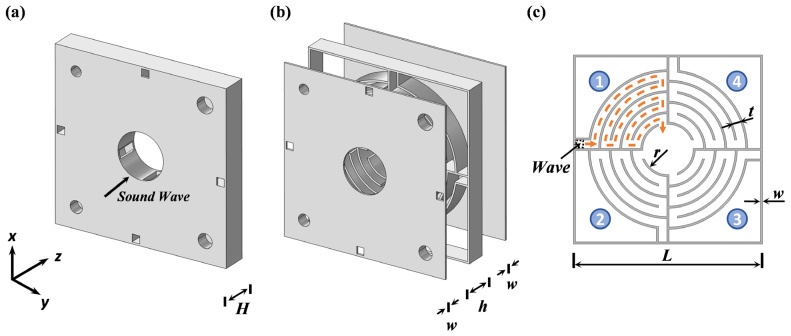
(**a**) Schematic diagram and (**b**) explosion view of VCMU. (**c**) Cross-sectional view of the middle layer.

**Figure 2 materials-18-02029-f002:**
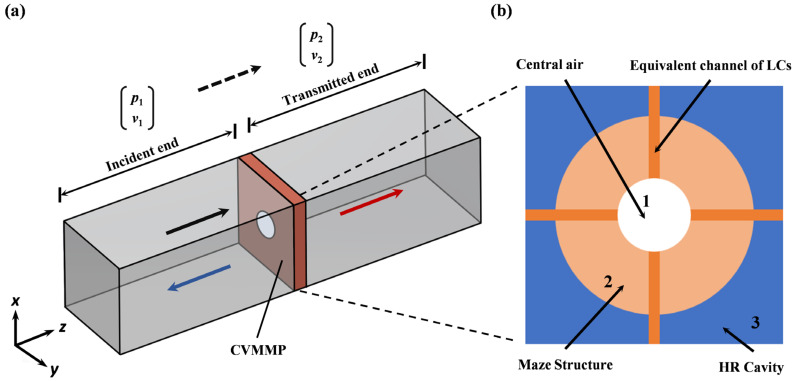
(**a**) Schematic of the structural waveguide. (**b**) Equivalent model of the structure, where the metamaterial is divided into three independent regions.

**Figure 3 materials-18-02029-f003:**
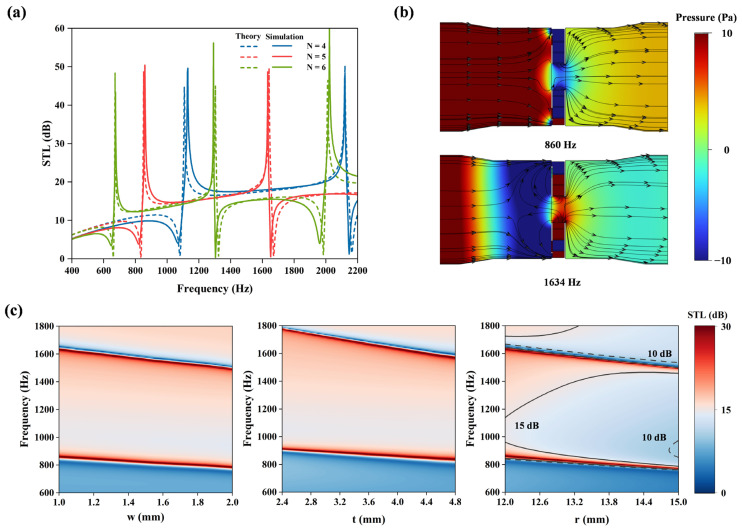
Theoretical and simulated numerical analysis of the Fano resonance characteristics: (**a**) STL spectra for varying spiral turns (*N* = 4, 5, 6) while maintaining other geometric parameters. (**b**) Sound pressure distribution and sound velocity streamlines at the Fano resonance frequency for *N* = 5. (**c**) Parametric STL analysis: Frequency-dependent STL for varying LC widths *t*, structural unit thickness *w*, and central runner radius *r*.

**Figure 4 materials-18-02029-f004:**
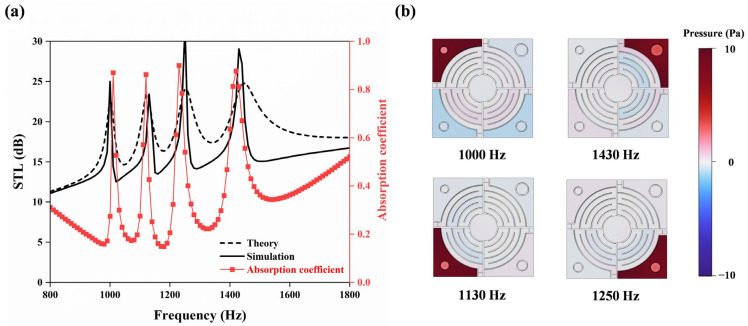
(**a**) Comparison of simulation and theoretical analysis results of the HR array and the sound absorption coefficient of HR array; (**b**) sound pressure diagrams at the corresponding resonance frequencies of each cavity in the HR array.

**Figure 5 materials-18-02029-f005:**
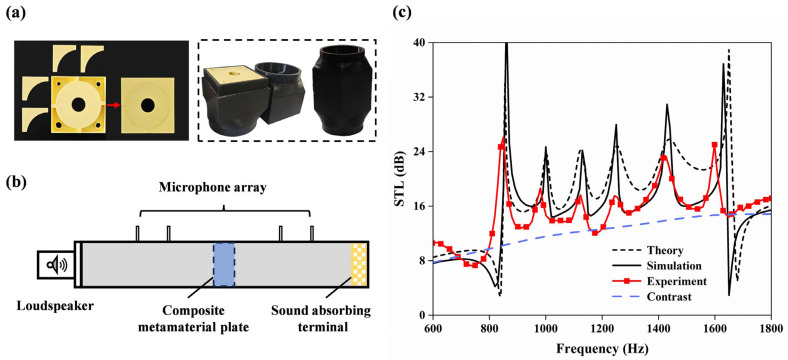
The configuration of (**a**) physical drawing of metamaterial unit and matching adapter tube and (**b**) measurement system. (**c**) Numerical analysis, simulation, and comparison of experiments.

**Figure 6 materials-18-02029-f006:**
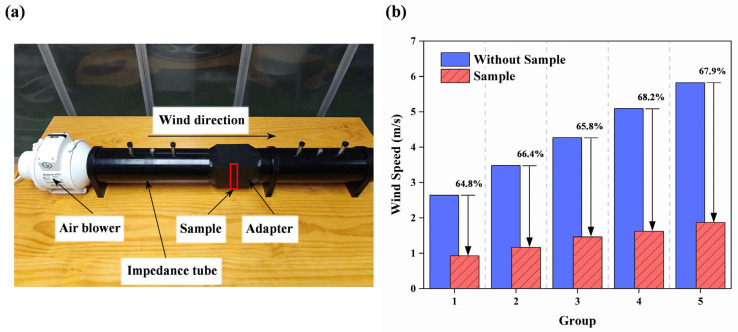
(**a**) Schematic diagram of ventilation measurement experimental device; (**b**) histograms of air flow rates with and without sample passage.

**Table 1 materials-18-02029-t001:** Detailed geometric parameters of VCMU.

Parameter Description	Symbol	Unit (mm)
Width of the square outer frame	*L*	90
Radius of the central circular channel	*r*	12
Single-layer thickness of the middle structure	*h*	11
Labyrinth channel and inlet width	*t*	4
Partition thickness	*w*	1

**Table 2 materials-18-02029-t002:** The structure parameters of HR cavity.

Unit *i*	*d_i_* (mm)	*V_i_* (mm^3^)	*f_i_* (Hz)
1	6.3	8143.2	1002
2	7.3	8088.5	1124
3	8.4	8012.5	1252
4	10.1	7846.2	1443

## Data Availability

The original contributions presented in the study are included in the article, further inquiries can be directed to the corresponding authors.

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
