# Peer review of "Deep-Subwavelength Composite Metamaterial Unit for Concurrent Ventilation and Broadband Acoustic Insulation"

_materials, 2025, doi:10.3390/ma18092029_

Round 1

Reviewer 1 Report

Comments and Suggestions for Authors

This manuscript presents a metamaterial unit integrating optimized labyrinth channels and Helmholtz resonators with a reduced thickness. It is suggested that a panel composed of units like this could achieve good sound insulation together with natural ventilation through the central flow channel of each unit.

The idea is of coupling sound insulation and natural ventilation very interesting, even if it already exists in the literature; however, there are several points that should be clarified and improved before the manuscript can be considered for publication. More details are given below.

The English/technical language needs to be revised because there are little flaws, unusual wordings, omitted words, etc.

The title may be misleading because the manuscript does not present a full “composite metamaterial panel” but only one basic unit to be replicated to get a composite panel.

I suggest improving it to: “Deep-subwavelength composite metamaterial unit for concurrent ventilation and broadband acoustic insulation”.

The literature review done in the Introduction must be improved.

For example, Ref. [33] is cited together with Ref. [34] as it were about sound insulation, while Ref. [33] is about sound absorption and ventilation is marginally addressed in it, if any.

Moreover, the authors write (lines 53-54) “Despite progress, existing designs predominantly deploy these mechanisms in stacked configurations [30–32]”. These references are not sufficient to cover the broad literature on simultaneous sound insulation and ventilation. The authors should read and cite more recent publications, e.g., http://dx.doi.org/10.1121/10.0020133, http://dx.doi.org/10.1121/10.0020133. A recent review (2025) on this topic can be found in https://doi.org/10.1016/j.buildenv.2025.112780. It can help the authors to discover more useful references.

In Section 2.1 there are several points to be improved.

The value of the end correction factor “gamma” is omitted (lines 111-112).

Formulas (7)-(9) are assumed without explanations. A couple of citations are not enough; these formulas should be derived analytically, or at least the main steps to arriving at them should be outlined.

The steps to go from formula (10) to formula (11) should be outlined.

In lines 119-120, it is written “Figure. 5(c) compares the theoretical predictions (short dashed line) and simulation results (solid line), showing excellent agreement in key frequency bands”. In my opinion, the agreement is not excellent but just fair, because the numerical results show much higher peaks than the theoretical calculation. Moreover, the experimental results show lower and larger peaks, as if additional resistive damping were present. I recommend avoiding excessive claiming here and expanding the comments on the discrepancies with the experimental results. They may be due to imperfect realization through additive manufacturing (see for example https://doi.org/10.1155/2019/7029143).

While the title, the abstract and the conclusion claim concurrent sound insulation and ventilation performances, the “good ventilating performance” (line 253) is not treated at all in this manuscript. Therefore, the study is incomplete. A Section must be added showing analytical/numerical and experimental evidence of the ventilation performance achievable with the studied metamaterial unit.

Finally, I remark that there is a big difference between a single metamaterial unit, measured in an impedance tube, and a full panel, e.g., with the size of a commercial window. The authors cannot omit to mention this point.

In conclusion, my opinion is that this work should be considerably improved before being considered for acceptance.

Comments on the Quality of English Language

The English/technical language needs to be revised because there are little flaws, unusual wordings, omitted words, etc.

Reviewer 2 Report

Comments and Suggestions for Authors

Please see document attached

Comments on the Quality of English Language

Although the English is correct, the manuscript could benefit from language proofing.

Round 2

Reviewer 1 Report

Comments and Suggestions for Authors

This is the second version of a manuscript presenting a metamaterial unit that integrates optimized labyrinth channels and Helmholtz resonators with a reduced thickness. It is suggested that a panel composed of units like this could achieve good sound insulation together with natural ventilation through the central flow channel of each unit.

In the first revision, I pointed out several issues to be clarified and improved.

  • The English/technical language has been revised, and now it is clear.

  • The title has been corrected as suggested.

  • The literature review in the Introduction has been improved, citing also some more references.

  • In Section 2.1, the value of the end correction factor “gamma” is now explicated.

  • The derivation of key formulas is now outlined, citing some references.

  • The comment to Figure. 5(c) is now correct.

  • A Section has been added showing experimentally the ventilation performance achievable with the studied metamaterial unit in comparison with a free flow in a tube.

However, some remaining issues have been detected, see below.

  • In line 151, a non-existing figure 2(c) is cited (?). Please correct.

  • In line 252, a reference for the formula given for Rd is omitted. Please add it.

  • In Figure 6(b), the word “sample” is misspelled as “simple”. Please correct.

  • The file named “materials-3546747-supplementary.pdf” is nothing else than another copy of the manuscript (?)

In my opinion, this work has been considerably improved and is nearly to be accepted for publication after correcting the remaining minor issues.

Reviewer 2 Report

Comments and Suggestions for Authors

Please, see document attached
